# Complications of Therapeutic Plasma Exchange in Pediatric Neuroimmune Disorders

**DOI:** 10.3390/children12111457

**Published:** 2025-10-27

**Authors:** Kathrin Eichinger, Markus Breu, Marleen Renken, Sandy Siegert, Elisa Hilz, Sarah Glatter, Dagmar Csaicsich, Michael Boehm, Christian Lechner, Barbara Kornek, Rainer Seidl

**Affiliations:** 1Division of Pediatric Pulmonology, Allergology and Endocrinology, Department of Pediatrics and Adolescent Medicine, Medical University of Vienna, 1090 Vienna, Austria; kathrin.eichinger@meduniwien.ac.at (K.E.);; 2Department of Neurology, Medical University of Vienna, 1090 Vienna, Austria; markus.breu@meduniwien.ac.at (M.B.); marleen.renken@gmail.com (M.R.);; 3Division of Neonatology, Pediatric Intensive Care Medicine and Neuropediatrics, Department of Pediatrics and Adolescent Medicine, Medical University of Vienna, 1090 Vienna, Austria; 4Division of Pediatric Nephrology, Department of Pediatrics and Adolescent Medicine, Medical University of Viena, 1090 Wien, Austria; 5Division of Pediatric Neurology, Department of Pediatrics I, Medical University of Innsbruck, 6020 Innsbruck, Austria; ch.lechner@i-med.ac.at

**Keywords:** therapeutic plasma exchange, pediatric neuro-immunological disorders, complications, clinical outcome

## Abstract

**Highlights:**

**What are the main findings?**
•Therapeutic plasma exchange (TPE) was safe and effective in children with neuroimmunological disorders, with 84% showing clinical improvement.•Complications were frequent but predominantly mild to moderate, reflecting comprehensive documentation of minor adverse events.

**What are the implications of the main findings?**
•TPE represents a valuable treatment option for pediatric neuroimmunological disorders, particularly in patients refractory to first-line treatment.•Standardized reporting of outcomes and complications is essential to improve comparability across pediatric TPE studies and to support the development of pediatric-specific clinical guidelines, as current practice is mainly based on adult data.

**Abstract:**

**Background:** Therapeutic plasma exchange (TPE) is an established treatment for immune-mediated neurological diseases in adults, but pediatric-specific data remain limited. This retrospective single-center study investigates the safety, complication profile, and clinical outcomes of TPE in children with pediatric neuroimmunological disorders (PNID). **Methods:** Medical records of pediatric patients who underwent TPE at the Medical University of Vienna between April 2006 and October 2022 were reviewed. Inclusion criteria required TPE initiation before the age of 18 years. Data collected included diagnoses, pre-TPE therapy, TPE characteristics, complications and clinical outcomes based on retrospective documentation. **Results:** A total of 53 patients (60% female, median age 13 years) were included and underwent 378 TPE procedures. Most common diagnoses were pediatric-onset multiple sclerosis (23%) and autoimmune encephalitis (19%). TPE was preceded by corticosteroids and/or intravenous immunoglobulin in 83% of patients. Complications occurred in 81% of patients and 23% of procedures and were predominantly rated mild to moderate (CTCAE I–II), including nausea, hypotension, and catheter-related issues. Severe complications (CTCAE III–IV) occurred in 11% of patients; no deaths were reported. Clinical improvement was documented in 84% of patients, with 42% showing significant improvement. **Conclusions:** TPE is a generally well-tolerated and effective treatment in PNID, with a high rate of clinical improvement and predominantly mild complications. The higher reported complication rate compared to other studies likely reflects more comprehensive documentation of minor adverse events. These findings support the use of TPE in PNID, particularly in cases refractory to first-line therapies. Standardized reporting of outcomes and complications is essential to improve comparability across studies and guide future clinical practice.

## 1. Introduction

Pediatric neuroimmune disorders (PNID) encompass a heterogeneous group of inflammatory conditions affecting the central and peripheral nervous systems, including Guillain–Barré syndrome (GBS), acquired demyelinating syndromes (ADS), autoimmune encephalitis (AE), and related antibody-mediated diseases. Comprehensive epidemiological data for PNID as a group are currently lacking, as most studies report incidence separately for individual disease entities. Reported annual incidence rates include approximately 0.5–1.5 per 100,000 children for GBS [1], 0.5 per 100,000 for pediatric-onset multiple sclerosis (POMS), and 1.5 per 100,000 for ADS [2]. The underlying etiology of PNID is predominantly immune-mediated, involving autoreactive B and T cells, antibodies targeting neural antigens, and cytokine-driven inflammation. Genetic susceptibility is increasingly recognized as an important cofactor, with variants in genes regulating immune responses or neuroinflammation (e.g., RANBP2 and other immune-regulatory genes) associated with pediatric autoimmune neuroinflammatory conditions [3,4]. Understanding these genetic contributions may have therapeutic implications, helping identify patients more likely to benefit from interventions such as TPE or requiring tailored treatment strategies.

Therapeutic plasma exchange (TPE) is an extracorporeal procedure that removes a patient’s plasma and replaces it with a substitute solution, such as albumin or fresh frozen plasma (FFP). It is used to eliminate pathogenic factors like autoantibodies and immune complexes, making it an essential therapeutic option for various immune-mediated diseases [5,6]. The American Society for Apheresis (ASFA) classifies TPE indications into different categories based on the strength of evidence and expert consensus [7]. For certain neuroimmunological diseases, such as GBS, N-methyl-D-aspartate receptor (NMDAR) encephalitis and myasthenic crisis, ASFA recommends TPE as a first-line therapy (category 1) [7,8,9]. In other conditions, including multiple sclerosis (MS), chronic inflammatory demyelinating polyneuropathy (CIDP), acute disseminated encephalomyelitis (ADEM) and neuromyelitis optica spectrum disorder (NMOSD), TPE is considered a second-line therapy (category 2) when first-line immunotherapy—such as high-dose corticosteroids or intravenous immunoglobulin (IVIG)—fails or provides insufficient clinical improvement [7,10,11,12]. While the ASFA guidelines provide a valuable framework for clinical decision-making, they are primarily based on data from adult patients, as pediatric-specific evidence remains limited [13]. For instance, an ASFA classification for TPE indications is not available for common pediatric neuroimmunological diagnoses such as myelin oligodendrocyte glycoprotein antibody-associated disease (MOGAD) [14], acute flaccid myelitis (AFM), and autoimmune encephalitis (AE), except for NMDAR encephalitis. However, in clinical practice, TPE is widely used for PNID [15]. With this retrospective single-center study, we aim to provide data on TPE-associated complications and clinical outcomes in PNID, contributing to a better understanding of its safety and effectiveness in this specific patient population.

## 2. Materials and Methods

### 2.1. Study Population

We retrospectively evaluated medical records of patients with PNID who were treated with TPE at the Medical University of Vienna, Austria, between 1 April 2006, and 1 October 2022. In addition to patients with PNID, we also included patients with herpes simplex virus (HSV) encephalitis and subacute sclerosing panencephalitis (SSPE), in whom an underlying immunological process was suspected. For inclusion into the study the first TPE had to be started before the age of 18 years. If patients underwent multiple TPE cycles during the study period, only the first cycle was included in the analysis. Evaluated variables included demographics, diagnoses, treatment before TPE, complications during TPE, anticoagulation during TPE, duration of single TPE session and outcome after TPE. Assessment of baseline comorbidities revealed that no patients had pre-existing renal or hepatic conditions that could have influenced treatment-related adverse events. In this study, complications were classified according to the Common Terminology Criteria for Adverse Events (CTCAE) established by the U.S. Department of Health and Human Services [16]. A detailed mapping of all recorded adverse events to their corresponding CTCAE grades is available in Appendix A. Adverse events were recorded if they occurred during the hospital stay in which the TPE cycle was administered. Longer-term or delayed complications occurring after discharge were not systematically captured due to the retrospective nature of the study. Due to the retrospective study design and limited detail in past documentation, CTCAE categories 1 and 2 were grouped as mild and moderate complications, and categories 3 and 4 as severe and life-threatening complications. The outcome was retrospectively assessed from clinical documentation based on neuropediatric examination, as standardized scoring was not used during follow-up visits. Outcome was evaluated based on the treating physicians’ documentation, taking into account neurological status and functional recovery during hospitalization, with particular focus on key neurological domains such as consciousness, cranial nerve function, muscle strength, gait ability, and coordination. As no standardized clinical scoring system (e.g., EDSS) was used, this represents a methodological limitation of our retrospective study. Therefore, the following scoring system was applied in this study: Score 1 for no clinical improvement, score 2 for minimal or slight improvement, score 3 for moderate improvement, defined as evident clinical improvement with residual neurological deficits or functional limitations, and score 4 for marked improvement, indicating near-complete or complete recovery (restitutio ad integrum). A score of 0 was given if symptoms recurred during the analysis period. All patients included in this study, or their caregivers provided informed consent for the use of their personal data. The study was approved by the institutional review board (Ethikkommission der Medizinischen Universität Wien, EK 1123/2015).

### 2.2. Statistical Analyses

The statistical analysis was conducted using Microsoft Excel, and the data were evaluated in a descriptive manner. According to distribution and sample size, continuous and discrete numerical variables were reported as median and interquartile range (IQR) and/or range where appropriate. Categorical variables were reported as frequency and percentage. Denominators of given frequency rates represent patients with available data.

## 3. Results

### 3.1. Patient Characteristics and ASFA Categories

In total, 53 patients met the inclusion criteria. Females accounted for 60% of the cohort (*n* = 32/53), and the median age at the start of TPE was 13 years (IQR = 9.5), ranging from 0.6 to 18 years. POMS was the most common diagnosis, representing 23% (*n* = 12/53) of cases. AE was diagnosed in 19% (*n* = 10/53) and included NMDAR encephalitis, autoimmune cerebellitis, rhomboencephalitis, and presumed autoimmune encephalitis without detectable specific autoantibodies. Among these, five patients (9%, *n* = 5/53) had NMDAR encephalitis. GBS and transverse myelitis (TM) each occurred in 9% (*n* = 5/53), followed by ADEM affecting 8% (*n* = 4/53) of patients. NMOSD, MOGAD, and HSV encephalitis were each diagnosed in 6% (*n* = 3/53), while myasthenia gravis (MG), isolated optic neuritis (ON) and opsoclonus-myoclonus syndrome (OMS) were each observed in 4% (*n* = 2/53). The rarest conditions, SSPE and AFM, each accounted for 2% (*n* = 1/53) of cases. Table 1 provides an overview of patient characteristics and diagnoses, along with their respective ASFA categories [7]. Diagnoses without an ASFA classification are listed as “not classified.” The flow of patient inclusion and data availability for the analyses is illustrated in Appendix A.

### 3.2. Treatment Before TPE

For 49 of the 53 patients, the therapy administered prior to the initiation of TPE was recorded. Treatment with high-dose glucocorticoids or IVIG were included in the analysis. The most common pretreatments were glucocorticoids and/or IVIG in 83% of patients (*n* = 44/53), while 9% of patients (*n* = 5/53) received no therapy before TPE. Combined therapy with both glucocorticoids and IVIG was administered to 25% of patients (*n* = 13/53). Therapy with glucocorticoids alone was given to 47% (*n* = 25/53), and therapy with IVIG alone was given to 11% of patients (*n* = 6/53). For 8% (*n* = 4/53) of patients, data on pretreatment was not available.

### 3.3. Characteristics of TPE Procedures

The total number of TPE procedures performed could be determined for 48 out of 53 patients, while the exact number of procedures for five patients remained unverifiable due to incomplete documentation. A total of 378 TPE procedures were performed in 48 children and adolescents during the analysis period, with a median of 7 procedures per patient (IQR = 2, range 5–17).

Time to treatment, defined as the interval between first symptom and the initiation of the first TPE, was available for 48 patients. The median time to first TPE was 17.5 days (IQR 27.8 days; range 3–195 days). The duration of individual TPE sessions was recorded for 44 patients, in 9 patients this data was incomplete. The median duration of a single session was 2.55 h (IQR = 0.9; range 0.75–4.5 h). The length of a TPE treatment cycle, defined as the interval between the first and last TPE session, was available for 48 patients. The median cycle length was 14 days (IQR 5.3; range 5–51 days). In clinical practice, the standard TPE regimen consisted of 7–10 sessions administered over approximately two weeks, with daily treatments during the initial sessions followed by procedures every other day. In patients with delayed or slow clinical improvement, session intervals were occasionally extended to weekly treatments.

Anticoagulation data were recorded for 50 patients. Citrate monotherapy was the most used anticoagulant, administered in 82% (*n* = 41/50) of cases, while heparin monotherapy was used in 18% (*n* = 9/50). One patient (2%, *n* = 1/50) received a combination of heparin and citrate. Heparin was last used in 2015, after which only citrate was employed, reflecting the transition from filter-based apheresis to centrifuge systems. The total exchange volume per TPE session was documented for 49 patients, with a median of 3150 mL (IQR = 2600; range 650–6300 mL). Additionally, the exchange volume per kilogram of body weight was recorded for 46 patients, showing a median of 63.8 mL/kg (IQR = 10.9; range 37–177.8 mL/kg).

### 3.4. Complications During TPE

Detailed documentation of single TPE procedures was available for 47 out of 53 patients, accounting for a total of 365 TPE sessions during the study period. Complications were recorded in 23% of all procedures (*n* = 85/365). Among the 47 patients, 81% (*n* = 38/47) experienced at least one complication during one or more TPE sessions, while 19% (*n* = 9/47) had no complications. Since patients could experience both grade I–II and grade III–IV complications according to CTCAE criteria [16], multiple entries per patient were possible. The majority (79%, *n* = 37/47) had grade I–II complications, while grade III–IV complications occurred in 11% of patients (*n* = 5/47). No grade V complications (death) were observed in this study (Table 2). Complications led to the termination of one or more procedures in 23% of the patients (*n* = 11/47). However, this did not indicate the discontinuation of the overall treatment plan but rather the cessation of individual procedures. Of the total 365 procedures, 4% (*n* = 14/365) were prematurely stopped due to complications.

### Type of Complications

Among the 47 patients who experienced complications, the most frequently observed adverse events were classified as mild to moderate (CTCAE I–II). Nausea and/or vomiting occurred in 32% of patients (*n* = 15/47), while 19% (*n* = 9/47) experienced at least one episode of hypotension. Other minor complications included pruritus (9%, *n* = 4/47), headache (6%, *n* = 3/47), and various transient symptoms such as abdominal pain, neck pain, increased serum calcium levels, anxiety, restlessness, tingling in the hands, and a transient increase in blood pressure, which were observed in 40% (*n* = 19/47) of patients. Catheter-related problems were reported in 43% of patients (*n* = 20/47), all of which were mild and non-infectious (CTCAE I–II). However, in 9% of patients (*n* = 4/47), catheter-related infections required medical treatment and were categorized as CTCAE III–IV. Additionally, self-limiting moderate shortness of breath (CTCAE III–IV) occurred in 4% of patients (*n* = 2/47). In one patient (2%, *n* = 1/47), the infection led to sepsis due to Staphylococcus aureus, which was successfully treated without further complications. Apart from this single case of sepsis, no life-threatening complications occurred during the study period, and no deaths (CTCAE V) were recorded. (Table 3).

### 3.5. Outcome

The outcome was analyzed in 45 out of 53 patients, while the records for the eight excluded patients were incomplete or missing. Outcome assessment was performed during hospitalization, prior to discharge, and therefore depended on the duration of the individual TPE cycle. In some patients, concurrent low-dose oral corticosteroids or intravenous immunoglobulin (IVIG) were administered following completion of TPE, primarily to restore immunoglobulin levels reduced during the procedure. A significant improvement (Score 4) was observed in 42% (*n* = 19/45) of patients, and 31% (*n* = 14/45) showed a moderate improvement (Score 3). A slight improvement (Score 2) was documented in 11% (*n* = 5/45) of patients, while no improvement (Score 1) was noted in one case (2%). A recurrence (Score 0) occurred in 13% (*n* = 6/45) of patients, five of whom had initially shown improvement. Overall, 84% (*n* = 38/45) of patients experienced clinical improvement.

## 4. Discussion

In our study, we evaluated the safety and complication profile of TPE in PNID. Across 365 TPE procedures performed in 47 patients, complications were observed in 23% of all procedures (*n* = 85/365) and in 81% of patients (*n* = 38/47). The majority of complications were classified as mild to moderate (CTCAE grade I–II). More severe complications (CTCAE grade III–IV) occurred in 11% of patients, and no deaths were reported. Complications led to early termination in 4% of procedures, but none resulted in discontinuation of the overall treatment regimen.

Comparing complication rates across studies on TPE in PNID, it becomes clear that direct comparisons are challenging due to differences in definitions of complications. In contrast to our study, which reported a complication rate of 81% of patients (*n* = 38/47), Eyre et al. encountered complications in 53% of patients (*n* = 31/58) undergoing TPE for PNID [17]. However, specific minor complications such as nausea, vomiting, headaches or catheter-related issues were not mentioned, which may reflect differences in reporting practices rather than actual absence of these events. Similarly, Akcay et al. described a complication rate of 70% (*n* = 7/10) in patients with central nervous system (CNS) demyelinating diseases [18]. Their study, however, included a smaller cohort and focused on catheter blockages as the only catheter-related complication. In contrast, our analysis reported a broader range of non-infectious catheter-related events that interfered with the procedure, including those requiring flushing or repositioning measures. Manguinao et al. observed a complication rate of 35% (*n* = 9/26) [19], and Yash et al. reported adverse events in 9% of patients (*n* = 3/32), without detailing minor complications [20]. Importantly, our study systematically recorded all adverse events, including mild and transient reactions such as nausea, pruritus, or short episodes of hypotension. This comprehensive documentation of minor complications likely contributed to the higher overall complication rate observed in our cohort compared with previous pediatric TPE studies and provides a more complete representation of the procedural safety profile.

While some studies report complication rates per patient [17,18,19,20,21], others present complication rates per procedure [21,22,23,24,25], reflecting differences in how complications are recorded and analyzed. For instance, Savransky et al. analyzed 65 patients with PNID undergoing a total of 524 TPE procedures and reported a complication rate of 5.9% per procedure (*n* = 31/524) [22]. In comparison, our study observed a higher complication rate of 23% per procedure (*n* = 85/365). However, despite the higher rate, the complications in our cohort were mostly graded mild to moderate. In contrast, Savransky et al. reported several serious catheter-related complications, including two cases of pneumothorax, one pulmonary air embolism, and one pulmonary thromboembolism that resulted in death. In our study, no deaths were observed. The only life-threatening event—a case of sepsis in one patient (*n* = 1/47)—resulted in full recovery. It is also worth noting that minor complications such as nausea, itching, or vomiting were not mentioned in Savransky et al. [22]. These differences may reflect varying documentation practices or focuses, underscoring the importance of standardized definitions and reporting methods when comparing complication rates across studies.

Several studies have included patients with both, PNID and other diagnoses, making direct comparisons with our study challenging. For instance, in the study by Lu et al. [25], pediatric patients with systemic lupus erythematosus, pesticide poisoning, rapidly progressive glomerulonephritis, severe purpura nephritis, and liver dysfunction were included, alongside those with PNID. The complication rate per procedure in Lu et al. was 12.7% (*n* = 152/1201), thus lower than the 23% observed in our study (*n* = 85/365). Lu et al. also analyzed the impact of diagnosis on complication rates, finding that NMDAR encephalitis had the highest overall complication rate and a notably higher incidence of itching and urticaria in these patients (*p* < 0.05), though no explanation for these findings was provided [25]. Due to limited patient numbers in subgroups, we refrained from similar statistical analyses.

Cortina et al. analyzed 18 pediatric patients treated with TPE, most of whom had renal diseases, only three patients (*n* = 3/18) had PNID. The complication rate was 10.6% per procedure (*n* = 30/280), with no reported deaths [23]. Similarly, Mazahir et al. included 46 pediatric patients primarily with renal and neurological conditions, reporting a complication rate of 7.1% per procedure (*n* = 21/293), with no deaths due to treatment complications, although two patients died from their underlying diseases [24].

The study with the highest complication rate in the literature reviewed is Michon et al.’s work, where complications occurred in 55% of procedures (*n* = 898/1632) and in 82% of patients (*n* = 149/186) [21]. This study included a wide variety of diagnoses, such as solid tumors, sickle cell anemia, and renal diseases, with only six patients having PNID (*n* = 6/186) [21]. While the 82% complication rate in Michon et al.’s study is similar to the 81% observed in our study, the comparison is limited due to the more severe complications reported, including two deaths, and the inclusion of different apheresis techniques and diagnoses.

The time from the first symptom to the first TPE was longer in four and shorter in only one of the comparable studies in which only PNID patients were included. In our study, the median of the time to treat was 17.5 days (range 3–198). The median time is longer in the study by Eyre et al., with 25.5 days (range 1–4575) [17], in the study by Manguinao et al. with 22 days (range 3–94) [19], and in the study by Savransky et al. with 23 days (range 3–186) [22]. Only in the study by Akcay et al. the time is shorter, with 12.5 days (range 4–67) [18]. ASFA generally recommends early initiation of TPE in neuroimmunological diseases, as it is associated with better clinical outcomes [7]. Supporting this, Llufriu et al. found that early initiation of TPE was a predictor for clinical improvement in CNS inflammatory demyelinating syndromes that did not respond to high-dose corticosteroids [2]. Similarly, Chevret et al. reviewed data on TPE in children and adults with GBS, showing that initiating TPE within seven days of the onset of symptoms led to better outcomes compared to starting treatment later [26].

In our cohort, 84% (*n* = 38/45) of patients showed an improvement in symptoms, ranging from mild to significant. This outcome was based on information from discharge letters provided by the treating physicians. Comparable studies reported improvement rates between 66% and 88% after TPE [17,19,20,22]. Some of these studies used official scales, such as the Visual Outcome Scale (VOS), Bladder Control Scale (BCS), Gait Scale (GS), Modified Rankin Scale (mRS), and Expanded Disability Status Scale (EDSS), while others, like ours, relied on the treating physicians’ documentation to assess clinical improvement after TPE. Due to the variety of scales used across studies, comparing outcomes is challenging. This variation underscores the need for more consistent outcome measures in future studies. It should be noted that, due to the retrospective and observational design of this study, a definite causal relationship between TPE and clinical improvement cannot be established. However, given the clinical context, TPE may have contributed substantially to the observed improvements. In our cohort, TPE was primarily administered to patients with severe disease who had shown little or no response to first-line treatments such as corticosteroids or IVIG. Therefore, it appears unlikely that spontaneous recovery or the natural disease course alone account for the observed clinical benefits. Nonetheless, as some patients continued or initiated additional immunotherapies during or shortly after TPE, synergistic effects of these treatments cannot be ruled out. This study has several additional limitations that should be considered. Due to its retrospective design, detailed information on overlapping autoimmune conditions and baseline comorbidities—such as pre-existing headache—was not consistently available, limiting our ability to assess their impact on treatment response or adverse events. Notably, all patients were free from underlying renal or hepatic disorders, suggesting that observed minor complications, such as pruritus or nausea, were likely attributable to TPE itself. Additionally, routine genetic testing was not performed during the study period; incorporating genetic profiling in future prospective studies could provide valuable insights into disease pathogenesis and help identify patients who may derive the greatest benefit from TPE. Future multicenter, prospective studies with standardized outcome measures and comprehensive assessment of comorbidities are warranted to further refine the safety and efficacy profile of TPE in pediatric neuroimmune disorders.

Strengths of this study include the relatively large number of procedures evaluated, detailed documentation of adverse events and responses, and a long observation period, which together contribute valuable data to the limited pediatric literature on TPE.

## 5. Conclusions

This study provides important insights into the safety and effectiveness of TPE for pediatric patients with neuroimmunological diseases. The observed complication rate was higher than in many other studies, likely due to our broader definition of complications. However, most of the complications were mild to moderate, no fatal events occurred, and only a small proportion of patients experienced CTCAE grade III–IV complications, supporting the overall tolerability of TPE.

The high clinical improvement rate of 84% underscores the potential efficacy of TPE, even in cases where first-line immunotherapies fail. Although differences in outcome measures across studies complicate direct comparisons, our findings are in line with existing literature supporting the use of TPE in PNID. Given the retrospective nature of the study, lack of standardized outcome scales, and the single-center design, further prospective, multicenter studies with unified outcome criteria are needed. Nonetheless, our results reinforce that TPE is an effective and generally well-tolerated treatment option for pediatric patients with neuroimmunological disorders—particularly when performed at specialized centers with experience in pediatric apheresis procedures.

## Figures and Tables

**Table 1 children-12-01457-t001:** Patient characteristics and ASFA categories (per patient, *n* = 53).

Diagnosis	ASFA Category	*n* (%)	Age in Years, Median (Range)	Sex M/F
All	-	53	13 (0.6–18)	21/32
POMS	II	12 (23)	15.5 (6.5–18)	4/8
AE	I/n.c.^1^	10 (19)	13 (3.5–17.5)	5/5
ADEM	II	4 (8)	9.5 (4.5–16.5)	2/2
TM	II	5 (9)	14.5 (9.5–15)	1/4
GBS	I	5 (9)	13 (2–17.5)	4/1
HSV encephalitis	n.c.	3 (6)	0.8 (0.6–0.8)	2/1
NMOSD	II	3 (6)	14 (10.5–17.5)	0/3
MOGAD	n.c.	3 (6)	6 (3.5–7)	2/1
ON	II	2 (4)	10 (4.5–16)	0/2
MG	I-II	2 (4)	12 (9.5–14)	1/1
OMS	III	2 (4)	1.5 (1.5–1.6)	0/2
SSPE	n.c.	1 (2)	14.5 (14.5)	0/1
AFM	n.c.	1 (2)	10 (10)	0/1

^1^ ASFA category I for NMDAR-encephalitis *n* = 5 (9%), ASFA category not classified for other autoimmune encephalitis; n.c.: not classified.

**Table 2 children-12-01457-t002:** CTCAE-graded complications per patient (*n* = 47).

	*n* = 47	%
No complications	9	19
≥1 complication	38	81
CTCAE grade I + II	37	79
CTCAE grade III + IV	5	11
CTCAE grade V	0	0

CTCAE = Common Terminology Criteria for Adverse Events.

**Table 3 children-12-01457-t003:** Number and type of complications per patient (*n* = 47).

Type of Complication	*n* = 47	%
Nausea/vomiting	15	32
Drop in blood pressure	9	19
Pruritus	4	9
Headache	3	6
Non-infectious catheter-associated complications	20	43
Infectious catheter-associated complication	4	9
Sepsis	1	2
Other complications	19	40

## Data Availability

The data presented in this study are available on request from the corresponding author due to privacy, legal, and ethical reasons.

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
