# Peer review of "Complications of Therapeutic Plasma Exchange in Pediatric Neuroimmune Disorders"

_children, 2025, doi:10.3390/children12111457_

Round 1
Reviewer 1 Report
Comments and Suggestions for Authors
In this paper, the authors perform a retrospective single-centre study examining the use of plasma exchange for neuroinflammatory disorders in children. The sample size is relatively large for pediatric neuroinflammatory disorders and robust data is provided around complications. I think insufficient data is provided to draw clear conclusions from an efficacy standpoint. Some additional comments/questions:
- Were there any parameters that defined the different levels of improvement (e.g. moderate versus marked)?
- While information on pre-treatment is described, were any patients receiving immune therapy concurrent with plasma exchange? What about additional immune therapies after TPE prior to the time outcome was assessed?
- Can the authors provide any data as to the interval between procedures during each patient’s cycle (e.g. daily exchanges, every other day, etc)?
- Did the authors look specifically at adverse events only during exchanges or was there a timeframe around exchanges where a sign/symptom would be classified as an adverse event?
- At what time frame after TPE was outcome assessed (I understand may not have been standardized, but a group average anyways)?
- I think it should be more clear in the discussion that it cannot be concluded to what degree TPE is driving the clinical improvement seen (versus effects of other immune therapies or disease natural history)
Author Response
Response to Reviewer 1
We sincerely thank the Reviewer for the thorough evaluation of our manuscript and the constructive comments, which have helped us improve the clarity and scientific rigor of our work. Below, we provide detailed responses to the Reviewers points.
Comment 1: Were there any parameters that defined the different levels of improvement (e.g., moderate versus marked)?
Response: We appreciate this important comment. No standardized clinical scoring system (such as the EDSS) was applied in this retrospective study. Improvement was evaluated descriptively based on the treating physicians’ documentation, focusing on key neurological domains such as consciousness, cranial nerve function (e.g., dysphagia, dysarthria, or ventilatory dependence), muscle strength, gait ability, and coordination. We acknowledge that this represents a methodological limitation, while the detailed documentation of complications remains a particular strength of our study (lines 127-137: Outcome was evaluated based on the treating physicians’ documentation, taking into account neurological status and functional recovery during hospitalization, with particular focus on key neurological domains such as consciousness, cranial nerve function, muscle strength, gait ability, and coordination. As no standardized clinical scoring system (e.g., EDSS) was used, this represents a methodological limitation of our retrospective study. Therefore, the following scoring system was applied in this study: Score 1 for no clinical improvement, score 2 for minimal or slight improvement, score 3 for moderate improvement, defined as evident clinical improvement with residual neurological deficits or functional limitations, and score 4 for marked improvement, indicating near complete or complete recovery (restitutio ad integrum). A score of 0 was given if symptoms recurred during the analysis period.).
Comment 2: While information on pre-treatment is described, were any patients receiving immune therapy concurrent with plasma exchange? What about additional immune therapies after TPE prior to the time outcome was assessed?
Response: We thank the Reviewer for this valuable question. The outcome was assessed during the hospital stay before discharge, depends on the duration of the TPE cycle. Some patients received concurrent low-dose oral corticosteroids or intravenous immunoglobulin (IVIG) following completion of TPE, primarily to replenish immunoglobulins removed during the procedure. This information has now been added and clarified in the Results section (lines 242-244: In some patients, concurrent low-dose oral corticosteroids or intravenous immunoglobulin (IVIG) were administered following completion of TPE, primarily to restore immunoglobulin levels reduced during the procedure.).
Comment 3: Can the authors provide any data as to the interval between procedures during each patient’s cycle (e.g., daily exchanges, every other day, etc.)?
Response: We thank the Reviewer for pointing this out. In clinical practice, the standard regimen consisted of 7–10 TPE sessions, typically administered daily for the first few sessions and then every other day. In patients with slower clinical improvement, treatment intervals were occasionally extended to weekly separations. This information has been added to the Results section for clarity (lines 187-193; The length of a TPE treatment cycle, defined as the interval between the first and last TPE session, was available for 48 patients. The median cycle length was 14 days (IQR 5.3; range 5–51 days). In clinical practice, the standard TPE regimen consisted of 7–10 sessions administered over approximately two weeks, with daily treatments during the initial sessions followed by procedures every other day. In patients with delayed or slow clinical improvement, session intervals were occasionally extended to weekly treatments.).
Comment 4: Did the authors look specifically at adverse events only during exchanges or was there a timeframe around exchanges where a sign/symptom would be classified as an adverse event?
Response: We appreciate this pertinent question. Adverse events were documented if they occurred during the TPE procedure or within the immediate periprocedural period, as reported in the medical records. Longer-term or delayed complications outside the TPE period were not systematically captured due to the retrospective nature of data collection. We have clarified this point in the Methods section (lines 119-122; Adverse events were recorded if they occurred during the hospital stay in which the TPE cycle was administered. Longer-term or delayed complications occurring after discharge were not systematically captured due to the retrospective nature of the study.).
Comment 5: At what time frame after TPE was outcome assessed (I understand may not have been standardized, but a group average anyways)?
Response: We agree that this is an important consideration. Due to the retrospective study design, the exact timing of outcome assessment could not be standardized across all patients. However, outcome was assessed during hospitalization, prior to discharge, and therefore depended on the duration of the individual TPE cycle. This has now been specified in the revised manuscript (lines 240-242; Outcome assessment was performed during hospitalization, prior to discharge, and therefore depended on the duration of the individual TPE cycle.).
Comment 6: I think it should be more clear in the discussion that it cannot be concluded to what degree TPE is driving the clinical improvement seen (versus effects of other immune therapies or disease natural history).
Response: We fully agree with this excellent point. We have revised the Discussion section to more clearly state that, given the retrospective nature of the study and concurrent or sequential use of other immunotherapies, it is not possible to determine to what extent clinical improvement can be directly attributed to TPE alone. This limitation is now explicitly discussed in the revised version (lines 340-348; It should be noted that, due to the retrospective and observational design of this study, a definite causal relationship between TPE and clinical improvement cannot be established. However, given the clinical context, TPE may have contributed substantially to the observed improvements. In our cohort, TPE was primarily administered to patients with severe disease who had shown little or no response to first-line treatments such as corticosteroids or IVIG. Therefore, it appears unlikely that spontaneous recovery or the natural disease course alone account for the observed clinical benefits. Nonetheless, as some patients continued or initiated additional immunotherapies during or shortly after TPE, synergistic effects of these treatments cannot be ruled out.).
Sincerely,
The Authors
Reviewer 2 Report
Comments and Suggestions for Authors
This study discuss the therapeutic plasma exchange in children with neuro-immunological disorders. This work contributes important real-world data on outcomes and complication profiles in a challenging patient population.
That said, several aspects require clarification and strengthening before the manuscript can be considered for publication:
-
Methods clarity: Please define your outcome scale more explicitly, describe how complications were classified using CTCAE, and ensure denominators (per-patient vs per-procedure) are consistently reported.
-
Internal consistency: Correct discrepancies (e.g., optic neuritis frequency in text vs Table 1) and provide a flow diagram to explain differences in sample size across analyses.
-
Results presentation: Add confidence intervals to key proportions, stratify outcomes/complications where possible (by diagnosis, timing, or anticoagulation era), and reconcile missing data.
-
Comparative context: Strengthen the discussion by directly comparing your complication rates with prior pediatric TPE reports, highlighting your more complete capture of “minor” events.
-
Tables and figures: Revise tables for clarity (with denominators in captions), correct typos, and consider adding a supplementary table showing CTCAE mapping.
Comments on the Quality of English Language
Language and style: The English is generally understandable but would benefit from copyediting to correct spelling, grammar, and terminology inconsistencies.
Author Response
Response to Reviewer 2
We sincerely thank the Reviewer for the constructive and insightful feedback. We greatly appreciate the time and effort the Reviewer dedicated to reviewing our manuscript and for providing comments that have helped us improve its clarity, methodological precision, and overall scientific quality. We have carefully addressed each point raised, as outlined below.
- Methods clarity:
Please define your outcome scale more explicitly, describe how complications were classified using CTCAE, and ensure denominators (per-patient vs per-procedure) are consistently reported.
We thank the Reviewer for this valuable comment. We have clarified the outcome assessment in the Methods section, specifying that clinical improvement was evaluated descriptively based on the treating physicians’ documentation, focusing on key neurological domains such as consciousness, cranial nerve function, muscle strength, gait ability, and coordination. As no standardized clinical scoring system (e.g., EDSS) was used, we acknowledge this as a limitation of our retrospective study (lines 127-132: Outcome was evaluated based on the treating physicians’ documentation, taking into account neurological status and functional recovery during hospitalization, with particular focus on key neurological domains such as consciousness, cranial nerve function, muscle strength, gait ability, and coordination. As no standardized clinical scoring system (e.g., EDSS) was used, this represents a methodological limitation of our retrospective study.). Conversely, the systematic documentation of complications represents a particular strength of our work. A supplementary table mapping all observed complications to the corresponding CTCAE grades has been added (line 119: A detailed mapping of all recorded adverse events to their corresponding CTCAE grades is available in Supplementary Table S1.), and denominators (per-patient vs per-procedure) are now explicitly indicated in the Results tables captions.
- Internal consistency:
Correct discrepancies (e.g., optic neuritis frequency in text vs Table 1) and provide a flow diagram to explain differences in sample size across analyses.
We thank the Reviewer for identifying these inconsistencies. The discrepancy in the reported frequency of optic neuritis has been corrected. Additionally, we have included a flow diagram as supplementary material illustrating patient inclusion and reasons for missing data across analyses to improve clarity and reproducibility (line 162: The flow of patient inclusion and data availability for the analyses is illustrated in Supplementary Figure S1.).
- Results presentation:
Add confidence intervals to key proportions, stratify outcomes/complications where possible (by diagnosis, timing, or anticoagulation era), and reconcile missing data.
We appreciate this excellent suggestion. After consultation with our statistics department, we decided that due to the retrospective, single-center design and the small, heterogeneous subgroups, our data are best presented descriptively rather than with inferential statistics. For future prospective studies, we plan to include stratified analyses and confidence intervals to allow for stronger statistical comparisons.
- Comparative context:
Strengthen the discussion by directly comparing your complication rates with prior pediatric TPE reports, highlighting your more complete capture of “minor” events.
We thank the reviewer for this valuable comment. Our Discussion already included a detailed comparison of complication rates with prior pediatric TPE studies, emphasizing differences in documentation practices and definitions of complications. To further enhance clarity, we have slightly refined this section to better highlight that our study systematically recorded minor and transient adverse events—such as mild hemodynamic changes and sensory symptoms—which may have been underreported in earlier studies. This comprehensive documentation likely contributed to the higher overall complication rate observed in our cohort (lines 272–277: Importantly, our study systematically recorded all adverse events, including mild and transient reactions such as nausea, pruritus, or short episodes of hypotension. This comprehensive documentation of minor complications likely contributed to the higher overall complication rate observed in our cohort compared with previous pediatric TPE studies and provides a more complete representation of the procedural safety profile.).
- Tables and figures:
Revise tables for clarity (with denominators in captions), correct typos, and consider adding a supplementary table showing CTCAE mapping.
We appreciate this helpful feedback. All tables and figures have been carefully revised for clarity, and denominators are now indicated in the captions. Typos have been corrected.
Sincerely,
The Authors
Reviewer 3 Report
Comments and Suggestions for Authors
I would like to thank the Editors for the opportunity to review this manuscript. The topic is timely and relevant, as pediatric-specific data on therapeutic plasma exchange (TPE) in neuroimmunological disorders remain limited. The study is well-structured, and the authors present interesting findings regarding both safety and efficacy outcomes. However, in its current form, the manuscript requires substantial revisions before it can be considered for publication.
Major Revisions
-
Introduction:
The introduction is relatively limited. I suggest expanding it by:-
Including epidemiological data on the prevalence of autoimmune disorders in the pediatric population.
-
Discussing whether there are available data on overlapping autoimmune conditions in children.
-
Adding information on whether genetic mutations have been associated with pediatric autoimmune neuroinflammatory diseases, as this could provide further context to disease pathogenesis and therapeutic implications.
-
-
Comorbidities:
It is unclear whether all patients included in the study had only one autoimmune comorbidity, or if some had multiple concomitant autoimmune conditions. If overlapping conditions were present, it would be important to clarify whether these patients responded similarly to TPE compared to patients with a single diagnosis, and whether multiple comorbidities benefited simultaneously from treatment. -
Other clinical comorbidities:
Please expand the discussion on other potential comorbidities (neurological or systemic) that might have influenced outcomes or the occurrence of adverse events. For example:-
Was headache already present as a pre-existing comorbidity?
-
Were there underlying renal or hepatic conditions that might have contributed to pruritus or other adverse effects?
Clarifying this point would strengthen the interpretation of both efficacy and safety data.
-
Minor Revisions
-
Acronyms HSV and SSPE appear first in the Materials and Methods section, but they are expanded only in the Results section. Please expand them upon first mention in the Materials and Methods.
-
Please expand the acronym CNS (central nervous system) at its first appearance in the text.
Author Response
Response to Reviewer 3
We would like to thank the Reviewer for their thoughtful and constructive comments. We greatly appreciate the time and effort taken to evaluate our manuscript. Below, we provide a detailed point-by-point response to each comment, including the corresponding changes made in the revised manuscript.
- Introduction:
Reviewer Comment: Expand introduction with epidemiological data, overlapping autoimmune conditions, and genetic mutations associated with pediatric autoimmune neuroinflammatory diseases.
Response: We appreciate this helpful feedback. We have expanded the Introduction by including additional epidemiological data on pediatric neuroimmune disorders (lines 64-79: Pediatric neuroimmune disorders (PNID) encompass a heterogeneous group of inflammatory conditions affecting the central and peripheral nervous systems, including Guillain–Barré syndrome (GBS), acquired demyelinating syndromes (ADS), autoimmune encephalitis (AE), and related antibody-mediated diseases. Comprehensive epidemiological data for PNID as a group are currently lacking, as most studies report incidence separately for individual disease entities. Reported annual incidence rates include approximately 0.5–1.5 per 100,000 children for GBS [1], 0.5 per 100,000 for pediatric-onset multiple sclerosis (POMS), and 1.5 per 100,000 for ADS [2]. The underlying etiology of PNID is predominantly immune-mediated, involving autoreactive B and T cells, antibodies targeting neural antigens, and cytokine-driven inflammation. Genetic susceptibility is increasingly recognized as an important cofactor, with variants in genes regulating immune responses or neuroinflammation (e.g., RANBP2 and other immune-regulatory genes) associated with pediatric autoimmune neuroinflammatory conditions. [3,4]. Understanding these genetic contributions may have therapeutic implications, helping identify patients more likely to benefit from interventions such as TPE or requiring tailored treatment strategies.).
We also added a paragraph discussing genetic susceptibility and its potential therapeutic implications, linking it to the rationale for TPE (lines 354-360: Additionally, routine genetic testing was not performed during the study period; incorporating genetic profiling in future prospective studies could provide valuable insights into disease pathogenesis and help identify patients who may derive the greatest benefit from TPE. Future multicenter, prospective studies with standardized outcome measures and comprehensive assessment of comorbidities are warranted to further refine the safety and efficacy profile of TPE in pediatric neuroimmune disorders.).
- Comorbidities:
Reviewer Comment: Clarify whether patients had multiple autoimmune comorbidities and how this affected outcomes.
Response: We appreciate this excellent suggestion. Due to the retrospective design, we do not have consistent information on overlapping autoimmune conditions. We now explicitly acknowledge this as a limitation and suggest it as a focus for future studies (lines 349-352: Due to its retrospective design, detailed information on overlapping autoimmune conditions and baseline comorbidities—such as pre-existing headache—was not consistently available, limiting our ability to assess their impact on treatment response or adverse events.).
- Other clinical comorbidities
Reviewer Comment: Please expand the discussion on other potential comorbidities (neurological or systemic) that might have influenced outcomes or the occurrence of adverse events, for example: pre-existing headache or underlying renal/hepatic conditions.
Response: We thank the Reviewer for this suggestion. Baseline data on headache were not consistently documented and are now acknowledged as a limitation in the Discussion (lines 349-352: Due to its retrospective design, detailed information on overlapping autoimmune conditions and baseline comorbidities—such as pre-existing headache—was not consistently available, limiting our ability to assess their impact on treatment response or adverse events.). We have also added a statement in the Methods indicating that no patients had pre-existing renal or hepatic conditions that could have influenced treatment-related adverse events (lines 114-115: Assessment of baseline comorbidities revealed that no patients had pre-existing renal or hepatic conditions that could have influenced treatment-related adverse events.). In the Discussion, we note that observed minor complications, such as pruritus or nausea, were most likely attributable to TPE itself rather than underlying comorbidities (lines 352-354: Notably, all patients were free from underlying renal or hepatic disorders, suggesting that observed minor complications, such as pruritus or nausea, were likely attributable to TPE itself.).
Minor Revisions
Reviewer Comment: Acronyms HSV and SSPE appear first in the Materials and Methods section, but they are expanded only in the Results section. Please expand them upon first mention in the Materials and Methods.
Response: We have revised the manuscript so that HSV (herpes simplex virus) and SSPE (subacute sclerosing panencephalitis) are expanded at their first mention in the Materials and Methods section.
Reviewer Comment: Please expand the acronym CNS (central nervous system) at its first appearance in the text.
Response: We have revised the manuscript so that CNS is expanded as “central nervous system” at its first occurrence.
We are grateful to the Reviewer for these valuable suggestions, which have substantially improved the clarity, completeness, and overall quality of the manuscript. We hope that the revisions satisfactorily address all comments and strengthen the presentation of our study.
Sincerely,
The authors
Round 2
Reviewer 2 Report
Comments and Suggestions for Authors
Well done
Reviewer 3 Report
Comments and Suggestions for Authors
Well done!